# The Role of Microbiota in Pancreatic Cancer

**DOI:** 10.3390/cancers15123143

**Published:** 2023-06-11

**Authors:** Valerio Papa, Tommaso Schepis, Gaetano Coppola, Michele Francesco Chiappetta, Livio Enrico Del Vecchio, Tommaso Rozera, Giuseppe Quero, Antonio Gasbarrini, Sergio Alfieri, Alfredo Papa

**Affiliations:** 1Department of Translational Medicine and Surgery, School of Medicine, Catholic University, 00168 Rome, Italy; valerio.papa@unicatt.it (V.P.); giuseppe.quero@unicatt.it (G.Q.); antonio.gasbarrini@unicatt.it (A.G.); sergio.alfieri@unicatt.it (S.A.); 2Digestive Surgery Unit, Fondazione Policlinico Universitario A. Gemelli, IRCCS, Largo Agostino Gemelli 8, 00168 Rome, Italy; 3Center for Diagnosis and Treatment of Digestive Diseases, CEMAD, Fondazione Policlinico Universitario A. Gemelli, IRCCS, Largo Agostino Gemelli 8, 00168 Rome, Italy; tommaso.schepis@gmail.com (T.S.); gcoppp@gmail.com (G.C.); michelefrancesco.chiappetta@gmail.com (M.F.C.); livioenrico.delvecchio@gmail.com (L.E.D.V.); tommaso.rozera@gmail.com (T.R.)

**Keywords:** gut microbiota, pancreatic cancer, periodontal disease, intrapancreatic microbiota, gut microbiota modulation

## Abstract

**Simple Summary:**

Pancreatic cancer is a devasting disease that has unfortunately proven very difficult to treat. Exploring new therapeutic options and getting an early diagnosis is crucial to improve the outcomes for those affected. Studying the intestinal, pancreatic, and oral microbiota offers exciting perspectives to understand the development of pancreatic cancer better. Particular expressions of the microbiota could help both for early diagnosis and for predicting the response to chemotherapy or immunotherapy. This area of research is very promising; therefore, further studies are needed to increase the available data.

**Abstract:**

Pancreatic cancer (PC) has an unfavorable prognosis with few effective therapeutic options. This has led researchers to investigate the possible links between microbiota and PC. A disrupted gut microbiome can lead to chronic inflammation, which is involved in the pathogenesis of PC. In addition, some bacterial strains can produce carcinogens that promote the growth of cancer cells. Research has also focused on pancreatic and oral microbiota. Changes in these microbiota can contribute to the development and progression of PC. Furthermore, patients with periodontal disease have an increased risk of developing PC. The potential use of microbiota as a prognostic marker or to predict patients’ responses to chemotherapy or immunotherapy is also being explored. Overall, the role of microbiota—including the gut, pancreatic, and oral microbiota—in PC is an active research area. Understanding these associations could lead to new diagnostic and therapeutic targets for this deadly disease.

## 1. Introduction

The gut microbiota is a complex microbial ecosystem that inhabits the human gastrointestinal tract. The human gut is colonized by 100 trillion microorganisms and over 1000 resident bacterial species [1]. It is now widely acknowledged that the gut microbiota significantly impacts human health and disease. Disruptions to the gut microbiota have been linked to various health issues, including metabolic and autoimmune disorders and certain types of cancer [2]. Recent research has also revealed that the gut microbiota is involved in critical physiological processes such as digestion, nutrient absorption, and immune system regulation [3]. Interestingly, even the pancreas, once considered a sterile organ, has been found to have a specific microbiota in non-pathological conditions [4]. The close anatomical connection between the pancreas and the gastrointestinal tract through the pancreatic duct is the basis for bidirectional communication between the gut microbiota and the pancreas, known as the pancreas–microbiota axis [4]. The mechanism whereby microbiota colonizes the pancreas has yet to be fully understood. The first proposed mechanism involves the migration of bacteria from the small intestine through the pancreatic duct system. Another proposed mechanism is the translocation of microbiota from the lower gastrointestinal system through the portal circulation or mesenteric lymphatic system [5,6,7]. While the mechanism of pancreatic microbiota colonization is still being investigated, it is clear that bacterial colonization may play a role in pancreatic health and disease. The pancreatic juice, a complex fluid that contains digestive enzymes, bicarbonates, and other substances that help neutralize the acidic chyme from the stomach, can impact the gut microbiome by creating an environment that is more or less hospitable to different types of microorganisms [8]. Recent studies indicate that changes in the gut microbiota may play a role in the development and progression of benign and malignant pancreatic diseases. In particular, the link between gut microbiota and pancreatic cancer (PC) is increasingly recognized [9]. Additionally, oral microbiota-specific patterns and periodontal disease have been associated with an increased risk of PC [10,11,12].

This study aims to review the current literature on the role of the gut, pancreatic, and oral microbiotas in the pathogenesis and progression of and response to PC therapy.

For this purpose, a literature search was conducted using PubMed for publications in the English language until March 2023. Original articles and reviews were identified using the following search terms: “pancreatic cancer” and “pancreatic ductal adenocarcinoma”, matched with each of the following keywords: “gut microbiota”, “dysbiosis”, “microbiota imbalance”, “oral microbiota”, “periodontal disease”, “fecal microbiota transplantation”, “antibiotics”, and “probiotics.” Additional articles were identified by reviewing the reference lists of selected pertinent articles. As a second step, we synthesize, analyze, and critically evaluate our sources to determine trends and patterns in theory, debates, conflicts, and gaps in the existing literature.

## 2. Microbiota and Pancreatic Cancer Pathogenesis

### 2.1. Gut Microbiota and PC

The gut microbiome may play a role in developing PC and other malignancies. Recent metagenomics analysis has shown that gut microbial communities in PC patients differ from healthy controls, with a significant reduction in α-diversity and an increase in β-diversity, as well as a higher presence of four bacterial species: *Veillonella parvula* and *atypica* and *Streptococcus anginosus* and *oralis*, coupled with a reduction in short-chain fatty acids (SCFA) producers, such as *Faecalibacterium prausnitzii* and *Clostridiales* spp. [13]. In addition, a study by Pushalkar et al., conducted on 32 patients with PC, showed a higher fecal representation of *Proteobacteria*, *Firmicutes*, *Bacteroidetes*, *Euryarchaeota*, and *Synergistetes* compared to healthy controls; a similar result was found for tissue tumors, analyzed by 16S rRNA [5,14]. Many species were highly represented in the fecal samples of PC patients, including *Alistipes shahii*, *Klebsiella pneumoniae*, *Clostridium symbiosum*, *Clostridium bolteae*, and *Streptococcus mutans* [15]. On the other hand, in the gut microbiota of PC patients, an increase in *Gammaproteobacteria* [16], *Synergistetes*, *Porphyromonas, Prevotella, Helicobacter pylori*, and *Bifidobacterium* [5,6,17] and a reduction in butyrate-producing bacteria were observed [18]. A significant decrease in SCFA, with the simultaneous higher activation of the mevalonate pathway, generates isoprenoids, such as geranylgeranyl and farnesyl pyrophosphate, which are involved in the activation of GTPases, such as Ras; these are responsible for carcinogenesis and PC progression [19].

Alterations in the gut microbiota can also provide inflammatory stimuli that favor the progression of chronic pancreatitis, a risk factor for pancreatic cancer [20,21,22]. Fungi in the gut microbiota, such as *Ascomycota*, *Basidiomycota*, and *Malassezia* [23], have also been linked to carcinogenesis through different species-dependent pathways, particularly the activation of mast cells and the release of pro-inflammatory cytokines mediated by the C3 complement-mannose-binding lectin pathway [24,25,26,27]. Beyond the impact on the metabolism of nutrients introduced with the diet, the gut microbiota influences the host’s physiology through their metabolites [28,29,30]. For instance, different bacterial strains are linked to inflammatory diseases, such as metabolic syndrome or obesity, as they stimulate the higher release and absorption of lipopolysaccharides (LPS) [31]. LPS is a component of the outer membrane of Gram-negative bacteria, and it plays a pivotal role in activating the host’s innate immune system [32]. Indeed, several studies found a more significant presence of LPS-producing bacteria in the gut microbiota and the tumor-associated microbiota of PC patients [33,34].

Furthermore, another bacterial metabolite that is possibly implicated in the pathogenesis of pancreatic cancer is trimethylamine N-oxide (TMAO). TMAO positively impacts long-term survival rates in patients with PC. A preclinical study conducted by Mirji et al. demonstrated its anti-inflammatory properties via activation of the type I interferon (IFN) pathway. Hence, bacteria expressing CutC, an enzyme that produces TMAO precursors, was significantly reduced in PC patients [35].

Gut dysbiosis is believed to be involved in pancreatic carcinogenesis by activating chronic inflammation. Specifically, LPS could trigger Toll-like receptor 4 (TLR4), leading to the inhibition of various tumor suppressor proteins, such as PTEN, pRb, MAP2K4, and p53, and the induction of HIF-1α and STAT3, which in turn directs the oncogenic sequences, promoting cell migration and epithelial–mesenchymal transition (EMT) [36,37,38]. In addition, LPS, interacting with NF-κB, MyD88, and AKT, upregulates the expression of programmed cell death ligand 1 (PD-L1) and promotes a reduction in the host’s immune responses via the apoptosis of tumor-infiltrating lymphocytes (TILs) [39]. Even if LPS may induce the long-term depletion of inflammatory cells, it can play the opposite role in the early stages, increasing the presence of local CD3+ and CD8+ T cells. Hence, the gut microbiota could cause PC via the induction of sustained inflammatory response with the runaway production of reactive oxygen species (ROS) and reactive nitrogen species (RNS) [40]. These reactive species can fragment DNA, disrupt cell membranes, and compromise the protein folding process, increasing oncogenes’ concentration. Another possible signaling pathway was proposed by Noureldein et al. in their preclinical model; crosstalk between gut microbiota and pancreatic cells also occurred through the mammalian target of the rapamycin (mTOR) pathway and was able to control cell growth, autophagy, and cytoskeletal organization [41]. Chronic inflammation, combined with higher levels of active oncogenes, such as Kras, and microbiota-induced barrier disruption, leads to pancreatic carcinogenesis [42,43]. Intriguingly, these signatures may predict different prognoses. A study by Riquelme et al. showed that long-term survivors (LST, who survived more than five years after surgery) had a higher presence of *Flavobacteria*, *Proteobacteria*, *Actinobacteria*, *Alphaproteobacteria*, and *Sphingobacteria* [44]. These different microbial signatures could also be used in diagnosing PC. In a recent multicenter study [45], shotgun metagenomic analysis of fecal microbiota showed *Veillonella*, *Akkermansia*, and *Streptococcus* enrichment using a panel of 27 microbial species, recording an accuracy of 0.84 of the area under the receiver operating characteristic curve (AUROC). Remarkably, a similar microbiological signature was found in healthy proton-pump-inhibitor (PPI) users, suggesting the possible role of PPI-associated gut dysbiosis in the pathogenesis of PC [13].

Summarizing the data available, it emerges that the dysbiosis of the intestinal microbiota and its metabolic products are implicated in pancreatic carcinogenesis. However, this hypothesis requires further studies to better characterize the strains and microbiome alterations involved.

### 2.2. Intrapancreatic Microbiota in PC

Intrapancreatic microbiome analysis has attracted increased attention in recent decades. Intestinal microbial communities can colonize pancreatic tissues through multiple pathways, such as lymphatic and portal circulation [4,11]. In healthy people, bacteria may provide the nutrients for pancreatic tissue in the case of food insufficiency. Dysbiosis could aggravate or induce inflammation in the pancreatic parenchyma. The intrapancreatic microbiota of PC patients, analyzed by Pushalkar and colleagues, revealed a higher presence of *Proteobacteria* (45%) and *Bacteroides* (31%), with an increased bacterial abundance compared to healthy controls [5]. A higher concentration of *Proteobacteria* in the samples from the PC patients was found in a study of 187 patients and in a survey conducted on 1526 samples from different tumor types [46,47,48]. On the other hand, a study conducted by Geller et al. suggested an enhanced ability to metabolize chemotherapies due to a higher abundance of *Gammaproteobacteria* [27,34]. Fluorescence in situ hybridization, applied to tumor-associated microbiota, recorded a decreased abundance of *Lactobacillus* spp. and an increased abundance of *Fusobacterium* spp., *Malassezia* spp., and *Firmicutes* [4,6,18,23,45,49,50]. Several species were linked to PC microbiota, an emblematic example of which is *Helicobacter (H.) pylori*, which nevertheless exhibits controversial properties. Even if some reports did not find any associations, other studies, such as that conducted by Risch and colleagues, recorded a higher risk of PC in patients with *H. pylori* seropositivity (OR: 2.78, 95% CI; 1.49–5.20, *p* = 0.0014) [51], mainly in CagA-negative samples. Intriguingly, *H. pylori* were previously found in the pancreatic tissues of 75% of patients with PC; these differed from normal pancreatic tissues, which were negative [52]. *H. pylori* is known to cause gastric cancer through its virulence factors, and it was hypothesized that one of these factors, in particular CagA, could impact pancreatic carcinogenesis via the activation of the Wnt/β-catenin pathway, disrupting the cell–cell adhesion and intracellular signaling that ultimately promote neoplastic transformation [53]. Similarly, pathogenic *Escherichia coli* in the pancreatic tissue could elicit pancreatic inflammation and damage. This strain could induce DNA damage with breaks and abnormal cross-links [53]. Under these conditions, it could cause chronic inflammation and subsequently enhance the probability of neoplastic transformation [43,54,55]. Furthermore, as gut microbiota, tumor-associated microbiota could predict long-term and short-term survival (LTS). Riquelme et al. demonstrated that LTS patients had a higher relative abundance of *Bacillus clausii Pseudoxanthomonas*, *Saccharopolyspora*, and *Streptomyces*, suggesting that they play a role as prognostic biomarkers [43,44]. Although other analyses did not observe any differences in the Shannon index between LTSs and short-term survivors (STSs), the Chao1 index was higher in LTSs, indicating higher diversity in patients with better prognoses [44,56]. Specifically, LTSs showed a higher relative abundance of *Desulfovibrio*, *Megasphaera*, *Flavobacterium*, *Enhydrobacter*, *Enterococcus*, *Sphingomonas*, and *Bradyrhizobium* [44,57,58]. In contrast, STSs samples had a higher presence of *Clostridium*, which frequently acts as an opportunistic pathogen, as well as *Neisseria*, *Actinomyces*, *Aggregatibacter*, and *Porphyromonas* [59].

In summary, the characterization of the pancreatic microbiota will provide important information from a pathogenetic and prognostic point of view with the aim that the best knowledge can lead to applications in clinical practice.

### 2.3. Periodontal Diseases and PC

Many epidemiologic studies worldwide have found an association between periodontal disease, tooth loss, and PC. A nationwide Swedish registry-based cohort study that followed more than 5 million individuals for a median of 7.2 years showed that individuals <50 years with mild dental inflammation, periodontitis, and root canal infections had a 35%, 56%, and 58% increased risk of developing PC compared with dentally healthy individuals at baseline, respectively. In the 50–70 year age range, only people with periodontitis exhibited a 20% elevated risk of PC, while no association was found among individuals aged 70 years and older [60]. Chang et al. also reported evidence of a correlation between periodontal disease and PC. The authors found a positive association between PC and periodontal disease among people aged 65 years or older (HR, 2.17; 95% CI, 1.03–4.57), but this association was not observed among people under 65 (HR = 0.83; 95% CI: 0.52–1.34) [12]. Gerlovin et al. utilized the oral health questionnaire from the Black Women’s Health Study (BWHS), which followed 59,000 African American women for an average of almost ten years, to evaluate the PC risk. They found that participants who reported adult tooth loss, regardless of reported periodontal disease, had a substantially increased risk of PC (HR 1.94, 95% CI 1.04–3.64). The association was stronger among patients with at least five extracted teeth and was observed mainly among nonsmokers with a lower baseline risk of PC. However, self-reported periodontal disease, irrespective of tooth loss, was not statistically significant, although the HRs were more outstanding than 1.5 [61]. In contrast, in an older prospective study on US male health professionals, the number of natural teeth at baseline was not associated with PC. In comparison, periodontal disease at baseline was associated with a 64% increased risk of PC. However, tooth loss during the four years before PC diagnosis was statistically significantly associated with increased PC risk after adjustment for baseline periodontal disease. When assessed jointly, periodontal disease and tooth loss over the past four years significantly increased the risk of PC, with a risk ratio of 2.71 (95% CI: 1.70, 4.32) compared to people with neither periodontal disease nor recent tooth loss. These results suggest that recent tooth loss could be a marker for the severity of periodontal disease and, indirectly, PC susceptibility [62]. According to a recent meta-analysis carried out by Maisonneuve et al., both periodontitis and edentulism were associated with a significantly increased risk of developing PC (+74% (RR = 1.74 95% CI: 1.41–2.15) and +54% (RR = 1.54 95% CI: 1.16–2.05, respectively), with no evidence of heterogeneity across studies and no evidence of publication bias. All eight studies included in the analysis provided risk estimates adjusted at least for age and sex. However, only six and four studies adjusted risk estimates for tobacco smoking and alcohol consumption, respectively [63]. A few existing reports study the relationship between periodontal disease and cancer mortality, with limited findings. To explore this relationship, Heikkilä et al. analyzed the Finnish healthcare register data [64]. Their research showed a strong positive association between periodontitis and PC mortality in an analysis adjusted for the effects of age, sex, calendar time, socioeconomic status, oral health, dental treatments, and diabetes [64]. However, the study was limited by the lack of information on smoking and alcohol use, known as independent PC risk factors [64]. Using the NHANES III data (*n* = 12,605), Ahn et al. reported a 2.3-fold increase in oro-digestive cancer mortality among those with moderate or severe periodontitis [65]. Nevertheless, although subjects with periodontal disease also tended to have excess risks for PC, no significance was reached in an adjusted analysis for age, sex, smoking, education, race/ethnicity, and BMI (RR = 4.56; 95% CI: 0.93–22.29) [65].

In summary, the above-reported data indicate a significantly positive association between periodontal disease and the risk of PC. However, the underlying biological mechanisms explaining this association have yet to be elucidated completely. A plausible hypothesis is that oral microbiota dysbiosis with a significant predominance of defined bacterial strains could determine periodontal disease and, in turn, an increased risk of PC [66]. However, this hypothesis requires further investigation.

### 2.4. The Oral Microbiota and PC

There is growing evidence suggesting that the oral microbiota is related to PC. However, the research that explores the direct association between the oral microbiome and PC risk remains limited; it is typically based on small cross-sectional studies with one-time sampling. In a small retrospective case–control study, Farrell et al. suggested that relative abundances of specific salivary bacteria could be biomarkers for early-stage PC [67]. Using HOMIM array profiling technology, the researchers assessed salivary samples from 10 PC cases and ten healthy controls, identifying 16 species/clusters significantly associated with the PC cases. Six were then confirmed by qPCR. After independent validation, *Neisseria elongata* and *Streptococcus mitis* (decreased in pancreatic cancer) were validated as biomarkers that yielded 96.4% sensitivity and 82.1% specificity for discriminating PC cases from healthy controls [67]. In addition, investigators observed a significant increase in *Granulicatella adiacens* and *S. mitis* levels in PC patients compared to patients with chronic pancreatitis [67]. Similarly, using high-throughput 16S rRNA sequencing, Torres et al. found a lower proportion of *Neisseria* in PC patient saliva than in healthy patients and other disease categories. However, this trend was insignificant [68]. Additionally, the authors identified a significantly higher ratio of *Leptotrichia* to *Porphyromonas* (LP ratio) in the saliva of patients with PC, suggesting that the LP ratio might be a possible PC biomarker. Finally, compared to data reported by Farrell et al., no difference in the *G. advances* and *S. mitis* levels was detected [68]. In a more extensive and prospective study, Fan et al. examined oral microbiota composition using direct bacterial DNA analysis of 361 people who went on to develop PC and from 371 matched controls [11]. This study demonstrated that carriage of the periodontal pathogens *Porphyromonas gingivalis* and *Aggregatibacter actinomycetem comitans* was associated with a 1.6-fold and 2.2-fold higher risk of PC, respectively [11]. Unlike previous studies that characterized bacterial DNA in PC patients’ saliva, in this study, saliva samples were taken up to 10 years before PC diagnosis, allowing researchers to determine the potential etiologic role of oral bacteria in PC. Moreover, the authors also determined that patients with a greater abundance of *Fusobacteria* and its genus *Leptotrichia* had a decreased risk of PC (OR = 0.94, 95% CI 0.89–0.99; OR = 0.87, 95% CI 0.79–0.95, respectively) [11]. To examine the relationship between the host immune response, oral microbiota, and PC risk, Michaud et al. used an immunoblot array to measure antibody titers against 25 oral bacteria strains in the prediagnosis blood samples of a large European cohort of patients [69]. They found that higher levels of ATTC 53,978 antibodies against *P. gingivalis* (>200 ng/mL) were related to a 2.1-fold higher risk of PC compared with those with undetectable or lower levels (95% CI 1.05 to 4.36) [69]. Alternatively, high levels of antibody titers against commensal, non-pathogenic oral bacteria, including *Fusobacteria species*, were associated with a decreased risk of PC (OR 0.55; 95% CI: 0.36, 0.83) [69]. Although antibody titers against oral bacteria depend on the individual’s specific immunocompetence level, the ATTC 53,978 level may indicate periodontal disease and PC susceptibility if confirmed. The direct characterization of both the oral microbiota and the immunological profile is required to improve our understanding of the association with PC.

In conclusion, the evidence available on the relationship between specific profiles of the oral microbiota and PC appears to be of extreme interest not only for identifying early biomarkers for diagnosis but, above all, because they could underlie pathogenic mechanisms that have not yet been elucidated.

## 3. Role of Microbiota in PC Progression

The underlying molecular mechanisms remain elusive despite compelling evidence supporting the link between gut microbiota dysbiosis and PC. The diverse mechanisms underlying the interplay between microorganisms and tumor pathogenesis and progression can be categorized into three principal functions: modulation of the host immune system, interaction with the host metabolism, and the direct activity of microbial toxins and metabolites (Figure 1) [70]. Several pro-inflammatory signaling pathways have been implicated in the development and progression of pancreatic ductal adenocarcinoma (PDAC), including those mediated by Ikß kinase 2, COX2, and interleukin-1α (IL-1α) [71,72]. Notably, the tumor microenvironment, particularly cancer-associated fibroblasts (CAFs), secretes IL-1α and perpetuates KRAS signaling, which drives PDAC [67,69]. Additionally, the activation of STAT3 by KRAS mutant cells can recruit myeloid cells that secrete interleukin-6 (IL-6) and contribute to disease progression [71,73]. The gut microbiome can modulate inflammation, as the PC microbiome can induce innate and adaptive immune responses that result in immune suppression and evasion [74]. Specifically, bacterial proteins such as LPS have been shown to exhibit a high affinity towards the toll-like receptors (TLR4 and TLR2) on immune cells, resulting in the recruitment of MyD88 or TRIF adaptor molecules. This activation triggers downstream signaling cascades, including MAPK and NF-κB, that produce various inflammatory cytokines (TNF-α; IL-1β; IL-8), ultimately resulting in cancer cell proliferation [75]. The same pathway has also been demonstrated to enhance the invasiveness of pancreatic ductal adenocarcinoma (PDAC) cells [76]. In a murine study, depletion of the gut microbiota was shown to reduce tumor volume and liver metastases following injection of KPC pancreatic cancer cells compared to non-depleted controls. This effect was associated with a significant increase in interferon gamma-producing T cells and a decrease in immunosuppressive IL17- and IL10-producing T cells within the tumor microenvironment [77]. Among metabolites, gut microbiota-derived SCFAs play a crucial role in PDAC carcinogenesis. SCFAs have been found to interact with G-protein coupled receptors (GPCRs), specifically GPCR41 and GPCR43, which are now referred to as free fatty acid receptors (FFAR) 3 and 2, respectively [78]. Vitro studies indicate that butyrate can inhibit PDAC cell proliferation and induce secretory differentiation [79]. Moreover, these compounds exhibit HDAC inhibitory activity, a property associated with their potential to exert anti-cancer, anti-inflammatory, and anti-fibrogenic effects [80]. On the other hand, it has been proven that pyruvate promotes the invasiveness of pancreatic ductal adenocarcinoma (PDAC) cells by inducing the epigenetic reprogramming of mesenchymal cells into cancer-associated fibroblasts [81].

Both liver-derived primary bile acids and gut-microbiota-derived secondary bile acids can exert carcinogenic and anticarcinogenic functions. Bile acids are known to interact with a diverse range of receptors, including the farnesoid-X-receptor (FXR), liver-X receptor (LXR), Takeda G-protein-coupled receptor 5/G-protein-coupled bile acid receptor (TGR5), constitutive androstane receptor (CAR), vitamin D receptor (VDR), and pregnane X receptor (PXR) [82]. After binding to TGR5, deoxycholic acid (DCA) leads to the activation of EGFR, mitogen-activated protein kinase, and STAT3 signaling pathways; this induces cell cycle progression in pancreatic adenocarcinoma cells [83]. Moreover, the induction of COX-2 expression has been demonstrated in the PC cell lines BxPC3 and SU86.86 following exposure to DCA and chenodeoxycholic acid (CDCA), thus leading to inflammation [84].

Polyamines are a class of organic compounds characterized by two or more amino groups. Those molecules have been shown to exhibit the capability to bind with diverse macromolecules, such as DNA, RNA, proteins, and acidic phospholipid 19 [85], thereby affecting the cell cycle and tumor proliferation. While cadaverine, putrescine, spermine, and spermidine are well-known examples of polyamines, it is now known that bacteria can produce a diverse range of other polyamines as well [86,87]. A study using metabolomics and metatranscriptomics to analyze the fecal microbiome in a murine pancreatic adenocarcinoma model revealed an upregulation of bacterial polyamine biosynthesis, which worsens with tumor progression [88]. The primary polyamines produced were putrescine, spermine, and spermidine. Consistent with these findings, elevated levels of serum polyamines were also observed in both pancreatic-adenocarcinoma-bearing mice and human patients.

Indoles, which are tryptophan derivatives, have been identified as ligands for the aryl hydrocarbon receptor (AHR) and the PXR receptor [89,90]. AHR activation plays a critical role in regulating the immune system [91,92], highlighting the potential importance of indoles in immune function. While there is a lack of direct experimental data on the effects of indole derivatives in pancreatic adenocarcinoma, one study demonstrates that selective AHR modulators, such as omeprazole and tranilast, can effectively modulate the invasive behavior of pancreatic adenocarcinoma cells [93].

Limited data exist on the direct role of microorganisms in developing and advancing PC. The involvement of porphyromonas peptidyl arginine deaminase (PPAD), which is synthesized by multiple *Porphyromonas species* [94], in promoting PC via P53 activity and KRAS mutation is a potential avenue of investigation. Moreover, the carcinogenic role of *H. pylori* in PC is controversial. In a study by Nilsson et al. [52], *H. pylori* was detected in the pancreatic tissue of individuals diagnosed with PDAC [95]. There are two main mechanisms through which *H. pylori* infection may lead to the onset and progression of PC. Still, neither feature the direct involvement of the microorganism mentioned above [96]: (1) The presence of *H. pylori* in patients is associated with a reduction in antral D cells, decreasing somatostatin secretion, and the subsequent stimulation of secretin secretion. These events could lead to pancreatic growth and an elevated risk of carcinogenesis. (2) The proliferation of *H. pylori* within the gastric corpus mucosa results in atrophic gastritis and decreased stomach acid secretion, subsequently leading to bacterial overgrowth and the heightened production of N-nitrosamines via bacterial catalysis. However, two recent meta-analyses provide conflicting data regarding the increased risk of PC in *H. pylori*-positive patients [97,98]. In conclusion, the role of gut microbiota in the progression of PC could be attributed to gut-derived metabolites (LPS, SCFA, secondary bile acids, and polyamines), which can elicit a long-standing inflammatory response, altering host immune response and increase the mutagen potential of carcinogens.

## 4. Microbiota Involvement in Therapy Response

As previously discussed, there is evidence of a complex interplay linking the microbiome to PC, conditioning the tumoral microenvironment through local inflammation and immune responses, and leading to a relevant impact on the natural history of the disease. In this scenario, from a broader point of view, the microbiome could represent a surrogate biomarker to better predict, detect, stratify, and eventually aid therapy of pancreatic disease. The peculiar features of this microbiome-conditioned microenvironment have also been shown to have a substantial impact on both immune and cytotoxic therapies; it may therefore have the potential to be used as an adjuvant tool to improve the standard of care. However, a deeper understanding of this field is needed, considering the burden of the pancreatic cancer survival rate despite the availability of a range of therapeutic tools [99].

### 4.1. The Microbiome as a Non-Invasive Biomarker in PDAC Management

Despite the persistent challenge of evaluating the microbiota’s composition as a biomarker in a broad range of diseases, there has been a concerted effort to explore its potential in oncological diseases, including PC.

Several datasets on the microbiota’s potential predictive and detecting value have been published. For example, a recent study conducted on genetically engineered pancreatic ductal adenocarcinoma (PDAC) murine models (with the spontaneous occurrence of PC) reported the increased abundance of *Proteobacteria* and *Firmicutes* in early-stage/non-detectable PDAC associated with an upregulation of the polyamine and nucleotide biosynthetic pathways, together with an elevated serum polyamine concentration, which the authors additionally verified in PDAC patients compared to healthy controls [88]. A study with similar objectives analyzed and compared the gut microbiota composition of small groups of patients with PC, precancerous lesions, non-alcoholic fatty liver disease, and healthy controls. However, a group of fourteen bacterial taxa could discriminate between PC patients and the healthy controls; this finding partially overlapped with the results from a previous Chinese cohort [33]. The identified signature needed to be more substantial considering the high inter-individual variability; moreover, no consistent data emerged when assessing the gut microbiota associated with precancerous pancreatic lesions, lowering the potential value of these data regarding early-detection strategies.

The analysis of the microbiome and the associated carcinogenic gene expression pathways in PDAC might improve our understanding and generate the possibility of better stratification, as was comprehensively documented in a recent study by Guo et al. [58]. In this study, the authors examined 62 resected PDACs and proved the correlation between microbial-related inflammation pathways and disease progression. Moreover, after performing a clustering (differentiation according to basal-like, hybrid, and classical subtypes) based on the molecular profile and consistent with disease outcomes, they observed significant differences in the harbored microbiota depending on the PDAC subtype, particularly when comparing the basal-like subtype (the most aggressive) to the others, with increased bacterial mass and richness and a higher abundance of specific bacterial genera (namely, *Acinetobacter*, *Pseudomonas*, and *Sphingopyxis*, which also proved effective in stratifying PDAC patients according to disease outcomes) [58]. Beyond the complex analysis conducted on the clear network correlation between the microbiome of basal-like tumors and the upregulation of specific gene programs related to inflammation, immune response, and carcinogenesis, the potential prognostic and stratifying value of the tumoral microbiota composition represents the most compelling element emerging from this engaging work [58].

As previously mentioned, the oral microbiome has also proven its relevance to this research area; Farrell et al. validated the combination of two bacteria of the salivary microbiota (*Neisseria elongata* and *Streptococcus mitis*, both decreased in pancreatic cancer) as a diagnostic biomarker of pancreatic cancer when matched with healthy controls (96.4% sensitivity and 82.1% specificity) [67]. In a similar setting, a significantly increased ratio of *Leptotrichia* to *Porphyromona* was observed in the salivary microbiota of patients with PC compared to both healthy control and patients with other comorbidities [64]; consistently with the observations of Farrell et al., *Neisseria* was also decreased. Kurihara et al. later identified a specific tongue-coating microbiome profile (including *Fusobacteria* and other three genera) capable of differentiating patients with pancreatic head cancer and liver cancer from healthy controls, thus paving the way for the potential application of a microbial signature as a biomarker of early disease, or even as a tool for prevention, considering the association of *Fusobacteria* with a worse disease course [49]. Mitsuhashi et al. observed that the presence of *Fusobacteria* in human specimens of pancreatic cancer (8.8% of 283 patients) correlated with a significant increase in cancer-specific mortality in this subpopulation [100], setting the stage for the possible utilization of this signature as a prognostic biomarker. Species of *Fusobacteria* can be found as part of the normal oral microbiota. However, they can eventually have an opportunistic effect and a further oncogenetic effect, particularly one related to colorectal carcinoma [101,102]. Moreover, infection with *Fusobacterium nucleatum* was reported to be associated with the colonization of the tumor tissue and the enhanced proliferation, migration, and invasion of the tumor cells in patients with PDAC [103], rendering it worthy of further assessment both as a biomarker and as a therapeutic target.

In conclusion, these data are only assessable and generalizable if properly evaluated in broader cohorts, with the possibility of removing the burden of confounding factors and concomitant comorbidities.

### 4.2. Cytotoxic and Immune Therapy

Currently, cytotoxic regimens are widely used as the first-line therapy in non-resectable PC, with the scientific community actively taking steps to improve the disease outcomes. In recent years, microbiota modulation is emerging as a surprising potential tool to assist the current therapies; however, it remains in the laboratory testing phase.

There is well-documented in vitro evidence regarding the role of specific microbial taxa in interfering with the cytotoxic effect of gemcitabine, a nucleoside analog widely used in PDAC, among other carcinomas. Several bacterial species, mainly belonging to *Gammaproteobacteria*, were proven to convert gemcitabine into its inactive metabolite 2′,2′-difluorodeoxyuridine, potentially through the long form of the bacterial enzyme cytidine deaminase (CDDL) [34], previously identified in the context of *Mycoplasma hyorhinis*-mediated resistance to gemcitabine [104,105]. Interestingly, when analyzing the tumor microbiota profiles of human PDAC surgical samples, *Gammaproteobacteria* were found in 51.7% of the evaluated samples; moreover, the cultured bacteria derived from the PDAC samples were proved to determine resistance to gemcitabine [34]. Similarly, a non-CDDL-mediated resistance to oxaliplatin was observed, as other bacterial species can confer non-CDD-mediated resistance to oxaliplatin [34]. It is, therefore, evident that the presence of specific bacterial species in the tumoral microenvironment confers a tenacious resistance to these drugs, even with high drug levels, and affects the non-target organs. On the other hand, specific antibiotics could reasonably increase the cytotoxic effect on the tumor. The use of antibiotics before or during gemcitabine-containing regimens was associated with better outcomes [106,107]; in particular, one retrospective, single-center, Japanese study recently reported increased progression-free survival and overall survival (OS) in 37 patients with advanced PC who received antibiotic therapy compared to 62 patients not exposed to antibiotics during systemic combination treatment with gemcitabine plus nab-paclitaxel (5.8 versus 2.7 months and 11.0 versus 8.4 months, respectively) [108].

Another recent retrospective cohort study included 3850 patients with primary metastatic PDAC treated with first-line gemcitabine or fluorouracil chemotherapy [109]. Patients who received antibiotics were matched based on propensity scores to those who did not receive antibiotics the month before or after beginning first-line chemotherapy.

The antibiotic receipt was associated with an 11% improvement in OS (hazard ratio [HR], 0.89; 95% CI, 0.83–0.96; *p* = 0.003) and a 16% improvement in cancer-specific survival (HR, 0.84; 95% CI, 0.77–0.92; *p* < 0.001) among patients treated with gemcitabine [109]. In contrast, there was no association between antibiotic receipt and OS (HR, 1.08; 95% CI, 0.90–1.29; *p* = 0.41) or cancer-specific survival (HR, 1.12; 95% CI, 0.90–1.36; *p* = 0.29) among patients treated with fluorouracil [109]. These results confirmed that antibiotics might modulate bacteria-mediated gemcitabine resistance and potentially improve PDAC outcomes.

Interestingly, fluoroquinolones, such as levofloxacin and moxifloxacin, exhibited proapoptotic and antiproliferative effects [110,111,112], with specific evidence related to PC [113], while they may still also act through immune response modulation [112]. However, antibiotics have shown mixed results when associated with cancer therapy [114], partially concerning the wide variability of the specific antibiotics evaluated, the cancer type, and the complexity of the microbial-immune interplay involved.

From a different perspective, biomodified bacteria can be a valuable vehicle for accessing the tumor microenvironment and promoting a relevant antitumoral effect in the context of a Trojan horse strategy; on this matter, Selvanesan et al. recently reported that the inoculation in mice models of PC with tetanus toxin-producing *Listeria monocytogenes* induced the activation of cytotoxic T cells and a consequent significant reduction in tumor mass after gemcitabine administration [115].

Currently, immune therapy is not a common approach in PDAC, as these tumors are highly resistant to this strategy [116]. In a recent preclinical study conducted on animal models of metastatic pancreatic cancer, the antibiotic-induced depletion of the gut microbiota decreased the tumor burden [77]. Moreover, in the tumor tissue, a relative reduction in IL17a+ pro-tumor T cells and an increase in the IFNγ+ subpopulation was observed, possibly linked to a potential increase in responses to checkpoint inhibitor blockers, which is typically ineffective in PC [117]. Interestingly, Rag1-knockout mice models lacking in mature B and T lymphocytes and anti-IL17a antibody-treated mice derived no benefits from antibiotics, ruling out the possibility that antibiotics play a direct beneficial role and suggesting an immune-mediated tumor attenuation. Additionally, a few forthcoming clinical trials are evaluating the association of checkpoint inhibitors and antibiotics in PC (neoadjuvant pembrolizumab + ciprofloxacin and metronidazole after a FOLFIRINOX regimen in surgically resectable pancreatic adenocarcinoma, NCT05462496).

In conclusion, the possibility of improving the efficacy of cytotoxic or immunotherapy through manipulating gut microbiota represents an intriguing opportunity but remains an area of open investigation.

## 5. Modulation of the Gut Microbiota

There is insufficient evidence on the modulation of the gut microbiota in cancers, mainly regarding other oncological diseases, focusing on the role of well-known tools such as probiotics [118], prebiotics, and SCFAs [14].

Among these, butyrate has previously shown a relevant in vitro antitumoral effect in PC cells, preventing tumor invasion via β4 integrin downregulation [119], as well as inhibiting proliferation and possibly conditioning cell differentiation and antigen expression [120]; however, no confirmation studies or recent data regarding the possibility of clinical applications are available.

On the other hand, fecal microbiota transplantation (FMT) represents a more intriguing opportunity for PC. It is one of the most effective interventions in modulating the gut microbiota composition, although its indication in clinical practice is still confined to treating *C. difficile* infection [121].

Riquelme et al. conducted a well-designed study on this topic, performing a human-to-mouse FMT from long-term survival (LTS) and short-term survival (STS) patients with advanced PC, as well as controls [44]. They explored the tumor microbiome’s composition, immune cell infiltration, and the tumor growth of PC specimens in patients and the recipient antibiotic-treated mice models [44]. Tumor growth was significantly reduced in mice who received FMT from LTS patients compared with both the healthy controls and recipients from STS patients, the latter showing these groups’ more significant disease burden. The evidence of a decreased antitumoral effect in the LTS-FMT mice after antibiotic treatment finally confirmed the pivotal role of the gut microbiota in tumor growth. Notably, the authors observed a significant increase in the densities of CD3+ and CD8+ T cells in the tumor specimens of LTS patients, which was positively correlated with the tumor microbiome diversity and specific bacterial genera (i.e., *Saccharopolyspora*, *Pseudoxanthomonas*, and *Streptomyces*) and was ultimately associated with better overall survival [44]. Similar results were achieved in LTS-FMT mice recipients.

In contrast, the transplantation of the same mouse model after neutralizing CD8+ cells showed no beneficial changes, thus confirming the immune-related interplay between a healthy gut microbiome and tumor growth. However, despite the excellent rooting of the FMT (40% of the microbiome of the human donor), less than 5% of the tumor microbiome was related to the donor [44]. On the other hand, there was a greater abundance of *Clostridiales* in mice who received FMT from STS patients, which was consistent with the original composition of the tumor microbiome in this cohort [44]. These data demonstrate that the gut microbiome participates in the tumor microenvironment with direct translocations, influencing the local immune response and microbiome’s composition. From a different perspective, FMT has already shown pronounced efficacy as a supportive treatment in patients experiencing the continuation of checkpoint inhibitor-related diarrhea conditioning treatment, with preliminary data on metastatic kidney carcinoma in a recent RCT from Ianiro et al. [122].

A phase I trial is ongoing to assess the safety, tolerability, and feasibility of FMT in resectable patients with PDAC (ClinicalTrials.gov Identifier: NCT04975217).

In addition, this study will evaluate the changes in the gut, oral, and tumor microbiome of PDAC patients after FMT and determine immunological/molecular changes in the tumor after FMT. Patients will undergo FMT during colonoscopy and successively receive FMT capsules orally once weekly for four weeks without disease progression or unacceptable toxicity. Patients then undergo standard-of-care resection of the tumor. After completion of the study treatment, patients will be followed up until 180 days after surgery. We hope to obtain encouraging data from this pilot study.

The reversal of general conditions predisposed to generating a poor microbiome profile is undoubtedly an additional option for indirectly modulating the gut microbiota and influencing the natural history of PC, as recently observed with the administration of metformin in a mouse PC model [123]. Moreover, the composition of the gut microbiome is also associated with cachexia, a well-known characteristic of PC disease associated with reduced survival [124]; on this matter, it was observed that the severity of muscular atrophy in mice models of acute leukemia was inversely related to the abundance of *Lactobacilli* in the gut. In contrast, the consequent administration of a specific consortium of *Lactobacilli* and prebiotics decreased cachexia, being at least partially associated with a positive effect on chronic systemic inflammation [125]. Improving the weight and the metabolic state alone in oncological diseases, particularly pancreatic tumors, effectively increased survival rates [126]. It is, therefore, possible to speculate that a specific profile of the microbiota’s composition might have an independent effect on PC by exerting an anti-inflammatory-related effect on muscular atrophy.

Finally, the role of nutrition and dietary supplements in modifying the risk of PC and its impact on PC progression should be underlined even if definitive proofs linking these beneficial effects to improving gut dysbiosis in humans are still uncertain.

A recent interventional trial comparing low dose n-3-fatty acids (FAs), either as fish oil (FO) or marine phospholipids (MPL) supplementation, resulted in similar and promising weight and appetite stabilization in PC patients with the improvement of their quality of life (QoL) [127]. In addition, consistent with experimental evidence, a case-control study including patients with PDAC and healthy controls found that high intake of dietary vitamin C or E mitigates the risk of meat-derived mutagen exposure and consequent 2-amino-3,4,8-trimethylimidazo[4,5-f]quinoxaline (PhIP)-related PDAC [128]. In conclusion, manipulating the gut microbiota represents an intriguing therapeutic option even if the evidence from clinical trials is still in its infancy, particularly as adjuvant or concomitant with conventional therapies.

## 6. Future Perspectives

The study of the relationship between microbiota and PC is still a relatively new field of investigation. Future research may provide insights into this link’s molecular mechanisms and help identify novel therapeutic approaches. One of the most promising topics is microbiota-based therapy, such as FMT, to alter the gut microbiome and improve outcomes. Additionally, future research may help identify specific microbial biomarkers that can be used for diagnosis, prognosis, and treatment selection. In conclusion, continued translational research could inform clinical practice and improve outcomes for patients with this challenging disease.

## 7. Conclusions

PC is a highly aggressive malignancy with a five-year survival rate of less than 10% [129]. Due to the lack of symptoms in its early stages and the disease’s aggressive nature, early diagnosis is challenging, making treatment difficult and decreasing the chances of survival. Treatment options for PC include surgery, chemotherapy, and radiation therapy, but these approaches are often limited by the disease’s aggressive nature and propensity to metastasize. However, given the poor prognosis associated with PC, further research is urgently needed to improve our understanding of the disease and develop new approaches for prevention, diagnosis, and treatment.

The role of the microbiota in developing and progressing PC is an emerging field of research with significant implications for diagnosing, treating, and preventing this deadly disease. While the mechanisms underpinning the link between the microbiota and PC remain poorly understood, the evidence suggests that alterations in the composition and function of the microbiota can influence the pathogenesis of PC via various mechanisms. The gut microbiota has been shown to play a potential role in the pathogenesis of PC (Figure 2). The reduction in α-diversity and increase in β-diversity—with a higher presence of *Veillonella parvula* and *atypica*, *Streptococcus anginosus* and *oral*, and a reduction in short-chain fatty acid producers (e.g., *Faecalibacterium prausnitzii* and *Clostridiales* spp.)—are described in patients with PC [13]. This gut dysbiosis can activate chronic inflammation and the production of reactive oxygen species (ROS) and reactive nitrogen species (RNS), leading to pancreatic carcinogenesis [40]. In addition to the gut microbiota, the oral microbiota also seems to play an essential role in PC pathogenesis. Indeed, patients with periodontitis, root canal infection, and dental inflammation are at higher risk for developing PC [12].

Moreover, the specific composition of the oral microbiota has been described as a potential pancreatic cancer biomarker [67]. The gut microbiota also plays a part in the progression of PC, modulating the host immune system (activating both the innate and adaptative immune systems), interacting with the host metabolism, and directly producing toxins and metabolites (e.g., polyamines and indoles) [70]. Recent studies have suggested that the gut microbiota may also influence the response to therapy in PC. Several potential mechanisms have been proposed to explain the link between gut microbiota and the therapy response in PC. For instance, gut bacteria can modulate the immune system and influence the metabolism and pharmacokinetics of chemotherapy drugs, impacting their efficacy. Despite the growing interest in the potential role of gut microbiota in therapy responses in PC, many questions still need to be answered. First, the precise mechanisms by which gut bacteria influence therapy responses in PC and the specific bacterial species involved remain unclear. Additionally, it is still being determined whether manipulating the gut microbiota could be a viable strategy for improving therapy responses in PC.

This paper has some limitations inherent in the type of publication, i.e., narrative review. In fact, given the subjective selection of bibliographic sources, even after sharing and discussion among the authors, some studies still need to be cited or adequately discussed. Furthermore, given the relative novelty of the topic, many of the data presented refer to in vitro or experimental animal studies. Therefore, they should be viewed cautiously regarding translating findings to humans.

In conclusion, the potential of microbiota-based interventions in preventing and treating PC is an exciting area of investigation that may ultimately lead to improved outcomes for patients with this devastating disease. However, further studies are needed better to elucidate the complex interactions between the microbiota and PC and to identify potential therapeutic targets.

## Figures and Tables

**Figure 1 cancers-15-03143-f001:**
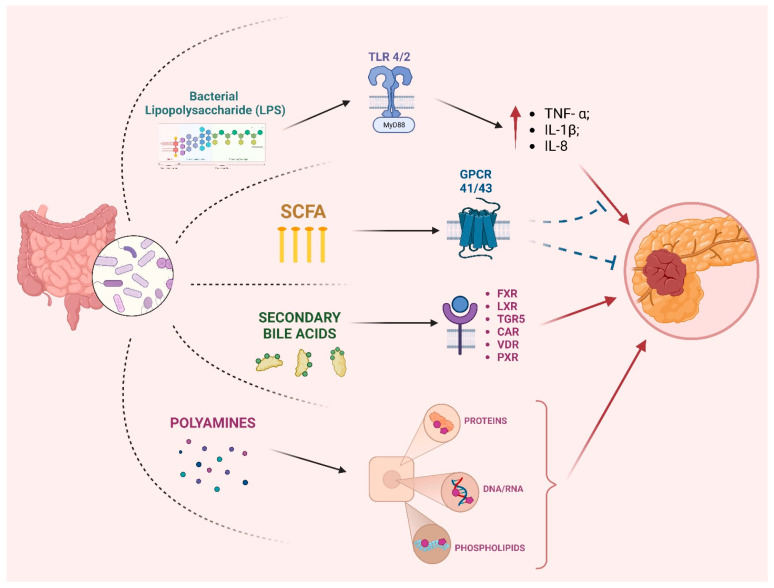
The influence of gut-microbiota-derived metabolites in the pathogenesis and progression of pancreatic cancer. Abbreviations: SCFA: short-chain fatty acids; GPCR: G-protein-coupled receptor; FXR: farnesoid-X receptor; LXR: liver-X receptor; TGR5: Takeda G-protein-coupled receptor 5/G-protein-coupled bile acid receptor; CAR: constitutive androstane receptor; VDR: vitamin D receptor; PXR: pregnane X receptor.

**Figure 2 cancers-15-03143-f002:**
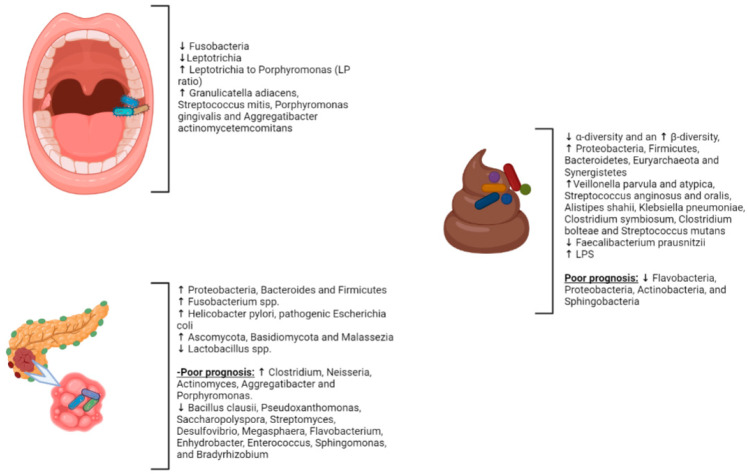
The most critical abnormalities found in the composition of the gut, oral, and pancreatic microbiota associated with pancreatic cancer and the alterations associated with a worse prognosis of pancreatic cancer. ↑, increased abundance; ↓, reduced abundance.

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
