# Peer review of "The Role of Microbiota in Pancreatic Cancer"

_cancers, 2023, doi:10.3390/cancers15123143_

Round 1

Reviewer 1 Report

1.     What are the search terms and search criteria adopted by the authors while collecting literature? Further, authors should also mention the search engines which are being explored during this literature analysis like Pubmed/Scopus/Web of Science, etc….

2.     In the introduction, authors have mentioned that “Additionally, oral microbiota and periodontal disease have been associated with an increased risk of PC.” It is devoid of proper citations.

3.     In section 2.1, unitalicized “and” in “Euryarchaeota, and Synergistetes”. Similar case has been observed at other places in the manuscript.

4.     There is no need to capitalize “L” of lipopolysaccharides in “absorption of Lipopolysaccharides (LPS)

5.     H. Pylori” should be italicized, and P of pylori should be in small case.

6.     Escherichia coli” should be italicized.

7.     It would be more appropriate to indicate key information of microbiota and PC in form of Tables.

8.     In section 2.3, authors discussed association of periodontal disease with PC. But unless authors will correlate periodontal diseases with oral microbiota, this section doesn’t seem fit the overall theme of the paper.

9.     Terminologies like in vitro, in vivo should be italicized.

10.   Role of Dietary Supplements should be discussed in managing gut microbiota that gradually impacting progression of PC.

11.   Section 5 is very brief.

12.   What are the limitations of this review article? It should be indicated in conclusion section.

13.   Before conclusion, authors should add a section “Future Perspectives”

Minor improvement in terms of English is required as indicated in my comments above.

Author Response

Point-by-point response to the reviewer’s comments.

The authors are grateful for the valuable comments of the Reviewers. We tried our best to address each observation in the following report. Corrections based on the comments were made, and a revised manuscript was attached. We hope our replies and modifications meet the Reviewers’ and Editor’s requirements.

Reviewer 1

1.     What are the search terms and search criteria adopted by the authors while collecting literature? Further, authors should also mention the search engines which are being explored during this literature analysis like Pubmed/Scopus/Web of Science, etc….

RE: We reported in the Introduction section the search terms and criteria adopted and the database used for collecting literature for this narrative review.

  1. In the introduction, authors have mentioned that “Additionally, oral microbiota and periodontal disease have been associated with an increased risk of PC.” It is devoid of proper citations.

RE: We added appropriate citations which reported the association between oral microbiota and periodontal disease with PC.

  1. In section 2.1, unitalicized “and” in “Euryarchaeota, and Synergistetes”. Similar case has been observed at other places in the manuscript.

RE: We have modified the text according to the reviewer’s comment.

  1. There is no need to capitalize “L” of lipopolysaccharides in “absorption of Lipopolysaccharides (LPS)”

RE: We have modified the text according to the reviewer’s comment.

  1. “H. Pylori” should be italicized, and P of pylori should be in small case.

RE: We have modified the text according to the reviewer’s comment.

  1. “Escherichia coli” should be italicized.

RE: We have modified the text according to the reviewer’s comment.

  1. It would be more appropriate to indicate key information of microbiota and PC in form of Tables.

RE: In figure 2 are summarised the key information regarding the alterations of gut, oral and pancreatic microbiota associated with PC. According to the Reviewer’s suggestion, we have explained this concept better in the legend of Figure 2. However, we think adding tables with similar data will be redundant.

  1. In section 2.3, authors discussed association of periodontal disease with PC. But unless authors will correlate periodontal diseases with oral microbiota, this section doesn’t seem fit the overall theme of the paper.

RE: Several epidemiological data reported a significant association between periodontal disease and PC. An intriguing pathogenetic link between periodontal disease and PC is oral microbiota dysbiosis. We explained better this intriguing hypothesis in the text.

  1. Terminologies like in vitro, in vivo should be italicized.

RE: We have modified the text according to the reviewer’s comment.

  1. Role of Dietary Supplements should be discussed in managing gut microbiota that gradually impacting progression of PC.

RE: We added a sub-paragraph regarding the role of Dietary Supplements in influencing gut microbiota and its possible impact on PC progression.

  1. Section 5 is very brief.

RE: We added more data (and references) to this section according to the Reviewer’s suggestion.

  1. What are the limitations of this review article? It should be indicated in conclusion section.

RE: We added in the conclusions the limitation of this review. In particular, given the narrative nature of the review and the subjective selection of references, we know they may have yet to cite some studies on the subject even after discussion among the authors. In addition, much of the data presented is from in vitro or experimental animal studies and, therefore, should be viewed cautiously when translating to humans.

  1. Before conclusion, authors should add a section “Future Perspectives”.

RE: We added a paragraph before the conclusions about future perspectives.

Reviewer 2 Report

The authors reviewed the current literature on the role of the gut, pancreatic and oral microbiota in the pathogenesis, progression and response to treatment of pancreatic cancer. The potential of microbiota-based interventions in the prevention and treatment of pancreatic cancer is an emerging area of research. Possible future research could lead to better outcomes of this patient group’s poor prognosis.

This review provides a comprehensive overview of a promising topic. The results of many different studies are described in detail and are presented in a well-organised manner. The two figures support the story well. However, it is not a very innovative article, given the many similar reviews already written on this topic which also use many of the same references. Suggestions for improvement:

-        Chapter 1 ‘Introduction’: Perhaps a search strategy with inclusion and exclusion criteria could be included in the introduction. It is now difficult to assess the completeness of the review. This may cause publication bias.

-          Chapter 2-5: As was done in chapter 4.1, adding a short conclusion after each (sub)chapter can provide even more clarity for the reader

-          Possible interesting recent studies to add:

o   DJ Fulop et al (2023), Association of Antibiotic Receipt with survival among patients with metastatic pancreatic ductal adenocarcinoma receiving chemotherapy. JAMA Netw Open. 2023 Mar 1;6(3):e234254.

o   Study Using Fecal Microbial Transplants in Patients With Pancreatic Cancer, NCT04975217 (Fecal Microbial Transplants for the Treatment of Pancreatic Cancer - Full Text View - ClinicalTrials.gov)

Author Response

Point-by-point response to the reviewer’s comments.

The authors are grateful for the valuable comments of the Reviewers. We tried our best to address each observation in the following report. Corrections based on the comments were made, and a revised manuscript was attached. We hope our replies and modifications meet the Reviewers’ and Editor’s requirements.

Reviewer 2

  • Chapter 1 ‘Introduction’: Perhaps a search strategy with inclusion and exclusion criteria could be included in the introduction. It is now difficult to assess the completeness of the review. This may cause publication bias.

RE: RE: We reported in the Introduction section the search terms and criteria adopted and the database used for collecting literature for this narrative review.

-    Chapter 2-5: As was done in chapter 4.1, adding a short conclusion after each (sub)chapter can provide even more clarity for the reader

RE: We added a short conclusion after each subchapter. 

-          Possible interesting recent studies to add:

o   DJ Fulop et al (2023), Association of Antibiotic Receipt with survival among patients with metastatic pancreatic ductal adenocarcinoma receiving chemotherapy. JAMA Netw Open. 2023 Mar 1;6(3):e234254.

  • Study Using Fecal Microbial Transplants in Patients With Pancreatic Cancer, NCT04975217 (Fecal Microbial Transplants for the Treatment of Pancreatic Cancer - Full Text View - ClinicalTrials.gov)

RE: We reported the most important findings of these studies in the text.